# Effect of Chitosan and Fish Gelatin Coatings on Preventing the Deterioration and Preserving the Quality of Fresh-Cut Apples

**DOI:** 10.3390/molecules24102008

**Published:** 2019-05-25

**Authors:** Yung-Shin Shyu, Guan-Wen Chen, Shao-Ching Chiang, Wen-Chieh Sung

**Affiliations:** 1Department of Baking Technology and Management, National Kaohsiung University of Hospitality and Tourism, No.1, Songhe Rd., Xiaogang Dist., Kaohsiung City 81271, Taiwan; tristar@mail.nkuht.edu.tw; 2Department of Food Science, National Taiwan Ocean University, 2 Pei-Ning Road, Keelung 20224, Taiwan; chengw@mail.ntou.edu.tw (G.-W.C.); hedy8313@gmail.com (S.-C.C.); 3Center of Excellence for the Oceans, National Taiwan Ocean University, Keelung 20224, Taiwan

**Keywords:** fresh-cut apple, chitosan, fish gelatin, quality

## Abstract

The effect of fish gelatin and chitosan coatings on the physicochemical characteristics of fresh-cut apples (*Malus pumila* Mill.), stored at 5 °C and 22 °C, was investigated. Chitosan provided an effective control for microbial growth, maintained firmness during 4 days of storage at room temperature (22 °C), and 12 days at refrigerator (5 °C). The results indicated that chitosan coating caused a significant decrease (*p* < 0.05) in the *L** value of cube color of cut apples. Fish gelatin–chitosan coatings mitigated the *L** value and decrease in hue angle of the cut apple samples, at cold storage. Experimental results showed that fish gelatin–chitosan and chitosan coatings, can be used to mitigate the formation of vitamin C, due to respiration, microbial growth, and weight loss at cold storage. Fish gelatin–chitosan coating might be a better combination for maintaining appearance and extending shelf-life of cut apples, compared to only chitosan coatings.

## 1. Introduction

Fresh-cut vegetables and fruits have attracted increasing attention because of convenience, health benefits, and their fresh-like character. Fresh-cut apples are more perishable and susceptible to deterioration, accompanied with microbial contamination, cut-surface browning, increased respiration, softening, and water loss (Rolle and Chrism, 1987; Qi et al., 2011) [1,2]. Fresh-cut apples are recommended to be stored at 5 °C, to reduce the rate of deterioration and biochemical reaction (Gil et al., 2006) [3]. It is not as easy to prevent enzymatic browning on fresh-cut fruits as is the case for whole fruits, as the cutting operation causes a release of polyphenol oxidase, which reacts with the phenolic compounds and forms a surface browning (Albanese et al., 2007; Pilon et al., 2014) [4,5]. Different apple varieties, and pre- and postharvest factors, such as concentration of gases in the storage atmosphere, also influence their susceptibility to browning (Hatoum et al., 2014) [6]. Edible coatings are useful strategies to mitigate physicochemical changes and reduce the respiration of fresh-cut apples (Zhou et al., 2008) [7]. Apple slices with chitosan-coating did not work very well due to water vapor barriers (Qi et al., 2011) [2], although edible coatings could reduce water loss, respiration, and oxidation reaction rates (Park, 1999; Xing et al., 2016) [8,9]. The oxidative reactions of phenolic compounds by polyphenol oxidase and its reaction products o-quinones in fresh-cut apples, make the brown apple slices unattractive (Qi et al., 2011) [2]. Edible coatings can maintain fruit quality, during storage, and can preserve their quality as efficiently as modified-atmosphere storage (Park, 1999; Olawuyi et al., 2019) [8,10]. Nevertheless, peeled oranges, bell peppers, and apples treated with different coatings, were not successful and were found to have even degraded their quality (Hagenmaier, 2005) [11].

Chitosan is a linear polymer consisting of randomly distributed β-(1-4)-linked *N*-acetyl-d-glucosamine and d-glucosamine, a partially deacetylated derivative of chitin, from the exoskeletons of crustacean shellfish, e.g., crabs and shrimps (Romanazzi et al., 2018) [12]. The degree of deacetylation in commercial chitosan, ranges from 65% to 99% (Muzzarelli and Rocchetti, 1985) [13]. It is biocompatible, biodegradable, and non-toxic to the human digestive system (Muzzarelli et al., 2012; Khalifa et al., 2017) [14,15]. Chitosan has also been widely used as an antimicrobial coating, due to its property of inhibiting the growth of fungi and many pathogenic bacteria (Romanazzi et al., 2002; Jovanovic et al., 2016; Valdes et al., 2017; Simonaitiene et al., 2015) [16,17,18,19]. Chitosan antimicrobial activity against bacteria, could be due to the polycationic nature of its molecule interactions and the formation of polyelectrolyte complexes, with acid polymers at the bacterial cell surface (Muzzarelli et al., 1990) [20]. Chitosan can produce alterations of membranes, which interact with the strongly electronegative microbial surface, leading to a change in metabolic disturbances and permeability, and eventually the killing-off of some fungi (Fang et al., 1994) [21]. Therefore, chitosan-based coatings have the potential to decrease oxygen levels, increase the shelf life of fresh vegetables and fruits, and inhibit the growth of microorganisms (Lazaridou and Biliaderis, 2002; Malinowska-Panczyk et al., 2015) [22,23].

Gelatin is a form of collagen hydrolyzed by thermal hydrolysis (Foegeding et al., 1996) [24] and exhibits unique gelling, colloidal properties, and a good ability to form coatings (Amiri et al., 2018; Samsi et al., 2019) [25,26]. Gelatin has long been widely and extensively used as a gelling and foaming agent in desserts, baking, jellied meats, candy, ice cream, and ice cream products (Wangtueai and Noomhorm, 2009) [27]. Commercial gelatin is primarily derived from pig skin, cattle hide, and demineralized cattle bone. Fish gelatin is an alternative for porcine gelatin and bovine gelatin, for certain religious uses, and it is a possible substitute for preventing contamination with bovine spongiform encephalopathy. Tilapia skin is a byproduct of the tilapia processing industry in Taiwan. Fish skin causes waste and pollution and it is processed to fish gelatin. However, so far the combined effect of fish gelatin–chitosan to inhibit browning and microbial growth, and to extend the shelf-life of fresh-cut apples has not been evaluated. The main objective of this research was to evaluate the effect of fish gelatin and chitosan edible coatings, on the quality of fresh-cut apples, during storage at 5 °C for 12 days and 22 °C for 4 days.

## 2. Results and Discussion

### 2.1. Weight Loss of Cut Apple Cubes

Weight loss occurred in all cut apples and the weight loss varied from 2.07% (fish-gelatin–chitosan coating) to 2.53% (control, water) after 12 days of storage at 5 °C. The apple cubes were exposed to the environment, which resulted in weight loss (Olivas et al., 2007) [28]. Weight loss was mainly due to the evaporation of water facilitated by a water vapor pressure gradient (Mohebbi et al., 2011) [29] and due to the loss of carbon, upon formation of CO_2_, during respiration (Kim et al., 2006) [30]. Coating did not significantly reduce the weight loss in cut apple cubes, although weight loss increased during storage, in this study, the weight loss (2.07%) of the fish gelatin–chitosan coated apple cubes was slightly lower (*p* > 0.05) after 12 days of storage, at 5 °C, compared to the control (2.53%) and other tested groups.

### 2.2. Microscopic Examination of Coating

As expected, higher coating thickness and lower weight loss was observed in this study (Figure 1). Fish gelatin–chitosan coating exhibited good water barrier characteristics, compared to the control (water). Micrographs of a cross-section of cut apple cubes coating obtained with a stereo-microscope, are shown in Figure 1. Chitosan, fish gelatin and fish gelatin–chitosan coatings covered the whole cut-surface of apple cubes and showed good adherence. Coating thickness were determined from the micrographs, estimating values of 37.12 μm ± 1.71 μm for chitosan, 57.04 μm ± 1.93 μm for fish gelatin, and 80.54 μm ± 5.45 μm for fish gelatin–chitosan, which were less than those of 0.5% gellan (155.75 μm ± 13.3 μm) and 2% alginate (132.45 μm ± 20.48 μm), reported by Rojas-Grau et al. (2007) [31]. However, their coatings also contained 0.6% glycerol (as plasticizer) and 1% *N*-acetylcysteine (as a hydrophilic antibrowning agent).

### 2.3. Color Changes for Storage

The enzymatic browning of cut apple cubes, during storage, was accompanied by a decrease in lightness (*L** value) (Table 1) and hue angle values (data not shown), and an increase in the browning index (Table 2). In comparison with fish-gelatin-coated cut apple cubes, chitosan and the fish gelatin–chitosan-coated samples and the control groups showed higher *L** values at 22 °C for 4 days (Table 1). Lightness (*L**) was the most indicative parameter associated with enzymatic browning for all tested samples of fresh cut apples (Moreira et al. 2015) [32], except for the chitosan-coated cut apple cubes. Although, at 22 °C on day 4 (Table 1), the color parameters, *L** and hue (h°), of both, the control and the fish-gelatin-coated cut apples exhibited significantly higher values than the chitosan-coated apple cubes, which had lower browning index values (*p* > 0.05) (Table 2). Our results confirmed that a 1.5% chitosan coating to the fresh-cut apple cubes could be used to extend the shelf life for more than 4 days, when stored at 22 °C. The combined fish gelatin–chitosan did not mitigate the quality deterioration at 22 °C. There were significant differences (*p* < 0.05) in the color values, between the control and the chitosan- or fish-gelatin-coated cut apple cubes, which had lower *L** values (Table 1) than the control, but there were no significant differences in the *L** values (*p* > 0.05) between the control and the fish-gelatin–chitosan-coated samples, when stored at 5 °C for 12 days. *L** values of the control and fish-gelatin–chitosan-coated cut apple cubes showed no significant decrease after 12 days of storage at 5 °C, nevertheless both *L** values of the chitosan and the fish-gelatin-coated cut apple cubes showed a significant decrease (Table 1), and their browning index values increased significantly (Table 2). The browning index was used as an indicator of browning, during storage (Olivas et al., 2005) [28]. Song et al. (2013) [33] observed that the browning index of apple slices increased rapidly for the first 4 h, during storage at 4 °C, for 16 days, and then leveled off. A drastic change in the color of fresh-cut apple samples was observed for the first 6 h, which was maintained thereafter (Qi and Hu, 2011) [34]. They proposed that browning of fresh-cut apples occurred during the early stage of storage. Figure 2 shows that the chitosan-coated cut apple cubes had a larger change than the other samples, in the browning color, during storage at 5 °C for 16 days. Cut apple cubes coated with fish-gelatin–chitosan was most effective in delaying the browning of apple cubes, during cold storage. Application of a combined fish-gelatin–chitosan coating is an effective approach to inhibit the enzymatic browning and the microbial growth of cut apple cubes, for 12 days at 5 °C (Table 2). *L** values of cut apple cubes decreased dramatically after 4 days at 22 °C, and decreased slightly after 12 days at 5 °C. A low temperature storage at 5 °C was better than a higher storage at 22 °C, to mitigate the enzymatic browning of fresh-cut apples. The retarding effect of the gelatin–chitosan coating was also confirmed by Qi et al. (2011) [2].

### 2.4. Firmness of Cut Apple Cubes

The firmness of cut apple cubes increased at the first 4 days storage at 5 °C, then decreased with increased storage time (Figure 3). The decrease of firmness has been reported in “Gala” and “Fuji” apples during storage (Pan and Tu, 2005) [35]. Fruit texture is closely related to cell composition and cell wall structure. Fruit softening is the consequence of disassembly of the middle lamella and primary cell wall structures (Jackman and Stanley, 1995) [36]. The data indicated that chitosan coating treatments mitigated firmness decrease better than the other groups. Coating treatments might maintain hardness by inhibiting water loss, due to the activities of pectin-degrading enzymes, and by reducing the rate of the metabolic processes during senescence (Zhou et al., 2008) [7]. Zhou et al. (2008) [7] reported that carboxymethyl chitosan coating were more efficient in reducing changes in the texture profile analysis (TPA) values of pear flesh than a sucrose/polyester base coating and the control sample, during a 4 °C cold storage. The maximum retention in firmness of Eksotika II papaya-fruit was obtained with 1.5% and 2.0% chitosan, at the end of storage at 12 °C, for 5 weeks (Ali et al., 2011) [37]. There was no significant (*p* > 0.05) difference between the firmness of the fresh-cut apple cubes treated with chitosan and the control group, up until the 4th day of storage, at 5 °C. There was a significant (*p* < 0.05) decrease in firmness in the control group at the end of storage, for 12 days at 5 °C. The maximum retention in firmness was obtained with the chitosan coating, showing a 0.5 N increase (Figure 3).

### 2.5. pH of the Cut Apple Cubes

A slight increase in pH values was observed after 12 days of storage at 5 °C (*p* > 0.05) (Table 3). However, a significant increase in the pH value (*p* < 0.05) of chitosan-coated cut apple cubes from pH 4.18 to pH 4.86 was measured, during 12 days of storage at 22 °C (Table 3), as compared to the other groups for which the pH of the cut apple cubes did not change dramatically, during storage at 22 °C for 4 days (Table 3). Polysaccharide-based multilayered coating did not change the pH of the papaya and the pH values for all fruits remained constant, by the end of 15 days of storage at 4 °C (Brasil et al., 2012) [38].

### 2.6. Total Soluble Solids (TSS) And Total Titratable Acidity (TTA) of Cut Apple Cubes

There was no significant difference among the tested groups and the control for the total soluble solids of cut apple cubes, during storage at 22 °C and 5 °C, which was similar to those reported by Bett et al. (2001) [39] and Song et al. (2013) [33] (Table 4).

Total titratable acidity of chitosan-coated cut apple cubes decreased significantly from 0.26% to 0.19% at 22 °C, for 4 days, as compared to the other groups, and it decreased to 0.14% after 12 days of storage at 22 °C. The titratable acid of all cut apple cubes in the tested and the control groups, decreased slightly from 0.25–0.26% to 0.20–0.23%, after 12 days of storage at 5 °C. There was no significant difference among the tested groups and the control during the same storage periods, irrespective of the storage temperature being 5 °C or 22 °C. These results were similar to those obtained by Albanese et al. (2007) [4] and Song et al. (2013) [33], who reported the titratable acidity to have decreased slightly. This phenomenon might be linked to a decrease in organic acids, due to an increase in respiration rate, following peeling and cutting.

### 2.7. Ascorbic Acid Content of Cut Apple Cubes

Ascorbic acid content in cut apple cubes increased significantly in the gelatin coating and the control groups, during 12 days of storage at 5 °C. Nevertheless, the ascorbic content of chitosan and fish-gelatin–chitosan-coated cut apple cubes maintained its levels (3.5 and 2.5 mg/100 g sample, respectively) after 12 days of cold storage. The ascorbic acid content increased in the whole papaya, during storage at 12 °C for 5 days (Ali et al., 2011) [37]. The chitosan coatings and fish-gelatin–chitosan coating seemed to have slowed down the synthesis of the ascorbic acid at cold storage. It might be because, both, the chitosan and the fish-gelatin–chitosan coatings provided a low oxygen permeability and delayed the vitamin loss during cold storage. Therefore, chitosan and fish-gelatin–chitosan worked as a modified atmosphere packing for delaying the synthesis of the ascorbic acid, in cut apple cubes, and it did not impair the cut apple’s respiration.

### 2.8. Microbiological Analysis of Stored Cut Apple Cubes

Effect of different coatings on the total aerobic bacteria, yeast, mold, and coliform of fresh-cut apple cubes during storage for 12 days, at 5 °C, are shown in Table 5. The population of total aerobic bacteria, yeast, mold, and coliform was not detected in chitosan and fish-gelatin–chitosan-coated cut apple cubes (Table 5). After 4 days at 22 °C, the population of the total aerobic bacteria in the control and fish-gelatin-coated samples was over 5 logCFU/g. The yeast, mold, and coliform populations exhibited a similar trend to that of the total aerobic bacteria (Table 5). The results were similar to *Aloe vera* gel-coated apple slices stored at 4 °C, for 16 days (Song et al., 2013) [33]. Some edible coating was effective in reducing bacteria as the coating substance formed a film which decreased the water activity on the surface of the fresh cut apple (Albanese et al., 2007) [4]. The growth of lactic acid bacteria, psychotrophs, mesophilic aerobes, yeasts, molds, and total coliforms, was inhibited by application of 1.5% chitosan-coated carrots stored at 10 °C for 15 days (Durango et al., 2006) [40]. The effect of chitosan on some fungi has been proposed by Fang et al. (1994) [21], due to the alterations in the functions of the cellular membrane. Chitosan coatings have a bactericidal effect on *Listeria monocytogenes* and it might be due to the positive charges of chitosan which interfered with the negatively-charged residues of the macromolecules at the Listeria cell surface, by competing with calcium for the electronegative sites on the membrane (Coma et al., 2002) [41]. Ouattara et al. (2000) [42] proposed that chitosan inhibited various spoilage bacteria through its capacity to inactivate various enzymes, bind water, and absorb nutrients that are normally used by bacteria. The results listed in Table 5 show that fish gelatin coating, alone, favored the development of aerobic bacteria, yeasts, molds, and coliform on cut apple cubes, compared to the control, indicating these microorganisms might have utilized protein and carbohydrate as a source of energy. Nevertheless, coating consisting 1.5% chitosan and 2% gelatin was the most efficient in controlling the microorgamisms, surface browning, increased respiration, and water loss, compared to 1.5% chitosan coating alone, which were for cut apple cubes stored at 5 °C.

### 2.9. Sensory Evaluation

The cut apple cubes stored at 22 °C for 4 days began to decay, except the chitosan-coated cut apple cubes. These cut apple cubes stored at 22 °C were, therefore, not presented to the panelists for sensory evaluation. The sensory evaluation results of the coated cut apple cubes stored at 5 °C for a duration of 12 days are shown in Table 6. The results revealed a significantly (*p* < 0.05) higher color score (5.77) of gelatin-coated cut apples than those of other groups. It might be because the gelatin could give the cut apples cubes a better appearance and color in light transmission and transparency. However, the control cut apple cubes attained the lowest score (2.93) in color, after 12 days of storage at 5 °C, which might be due to the browning color that these cut apple cubes looked like they started to decompose. The cut apple cubes coated with chitosan and fish-gelatin–chitosan obtained lower odor score, which might be because chitosan had to dissolve in 1% acetic acid and then adjust to a pH 6.0. The acetic acid smell could be detected by the panelists, for the odor and flavor attributes, which downgraded the scores of chitosan- and fish-gelatin-coated cut apple cubes, all the way during the 12 days of storage (Table 6). Nevertheless, the texture scores of the chitosan and the fish-gelatin–chitosan-coated cut apple cubes stored at 5 °C for 12 days was deemed more acceptable than those of fish–gelatin-coated and the control sample. This higher texture score of the chitosan and the fish-gelatin–chitosan-coated cut apple cubes stored at 5 °C for 12 days, was linked to a higher hardness of cut apple cubes (Figure 3). Small differences in overall quality among the samples were reported by the panelists, who rated the overall quality of the coated cut apple cubes as ‘slightly dislike’ to ‘neither like nor dislike’ after 12 days storage at 5 °C (Table 6).

## 3. Materials and Methods

### 3.1. Materials

Apples (Fuji #4129) were purchased from the local Ren Ai traditional market (Keelung, Taiwan) at commercial maturity and stored at 4 °C until processing. The 310–375 kDa high molecular weight chitosan (HC), acetic acid, metaphosphoric acid, and 2,6-dichloroindophenol were purchased from Sigma-Aldrich (St. Louis, MO, USA). Tilapia skin gelatin 80 mesh (200 Bloom) was purchased from Jellice Pioneer Provate Limited Taiwan Branch (Pingtung, Taiwan). All chemical reagents used were of analytical grade.

### 3.2. Preparation of Chitosan-Gelatin Coating

Coating solutions were prepared according to the method described by Seyed et al. (2013) [43]. Chitosan solution was prepared with 1.5% (*w*/*v*) chitosan in 1% (*v*/*v*) acetic acid, stirred for 24 h at room temperature (22 °C). The pH value of chitosan solution was adjusted to 6.0, by adding 0.1 N sodium hydroxide, and was then filtered through a micron wire mesh to remove impurities. Glycerol (0.3 g/g chitosan) was added as a plasticizer and the solution was warmed and stirred at 45 °C, for 15 min. Gelatin solution (4%, *w*/*v*) was prepared by dissolving 4 g tilapia skin gelatin in 100 mL distilled water, for 30 min, and then heated at 45 °C for 30 min, under continuous stirring. 1.5% (*w*/*v*) chitosan–gelatin solution with ratio of chitosan to 2% (*w*/*v*) gelatin (1:1) was prepared and heated at 45 °C, for 30 min, under continuous stirring. Glycerol (0.3 g/g gelatin) was added and warmed, same as the above procedure, for the chitosan solution and the solution containing both chitosan and gelatin.

Apples were washed, rinsed, and dried prior to the peeling operations. The apple was peeled, cored, and diced into 10 mm × 10 mm × 10 mm cubes. Apple dices were dipped for 2 min into sterilized distilled water (control), chitosan, chitosan–fish-gelatin, or a fish gelatin solution. The excess coating solution was allowed to drip off for 1 min, before being placed into polypropylene containers (600 cm^3^). The containers were stored at 22 °C and tested at intervals of 1 day, for the first 4 days of storage. Containers stored at 5 °C and 22 °C were also tested at intervals of 4 days during the 12 days of storage. 

### 3.3. Weight Loss of Apple Cubes

The percent of weight loss was recorded as the weight loss throughout the storage duration. The weight of each sample was record on day 0 and, throughout storage, at 5 °C and 22 °C, using a digital balance. The results were reported as an average of 15 apple cubes per treatment, as described by Brasil et al. (2012) [38], with slight modification, g/100 g weight loss = ((initial weight) − (final weight)/(initial weight)) × 100.

### 3.4. Color Measurement

The surface color of the cut apple was examined with a spectrocolorimeter (TC-1800 MKII, Tokyo, Japan) using *L**(lightness), *a**(redness/greenness), and *b**(yellowness/blueness) color scale. Both, a white tile and black cup were examined before the test to standardize the spectrocolorimeter. Three color measurements were recorded for each sample and duplicate determinations were recorded for each treatment. h° was referred to as color, which was the angle of tangent^−1^
*b**/*a**, where 0° = red purple, 90° = yellow, 180° = blue-green, and 270° = blue (Ali et al., 2011) [37]. The results were also reported as a browning index (BI). BI was calculated as follows (Song et al. 2013) [33]:BI = 100(x − 0.31)/0.172(1)
where x = *a** + 1.75*L** + *a** − 0.312*b**

### 3.5. Firmness of Cut Apple Cubes

Cut apple cube firmness was evaluated using a TA-XT2 Texture Analyzer (Stable Micro Systems Ltd., Haslemere, UK), with a load cell of 2 kg. The samples of firmness were tested under the plunger of the instrument, equipped with a 4 mm diameter probe, to penetrate into apple cubes. These were then randomly withdrawn from each container and placed perpendicular to the probe, so as to penetrate the center of the cut apple cubes. The firmness was expressed as the maximum force (g) of a plunger, with a speed of 5 mm/s, required to penetrate into the cube to a depth of 5 mm (Moreira et al., 2015) [32].

### 3.6. pH of Cut Apple Cubes

The pH of the apple cubes was measured using the AOAC method 981.12 (AOAC 1998) [44] with a digital pH meter (pH510m Eutech Instruments Pte Ltd., Ayer Rajah Crescent, Singapore) that was calibrated with buffer solutions at pH 4.0 and 7.0.

### 3.7. Total Soluble Solids (TSS) and Total Titratable Acidity (TTA) of the Cut Apple Cubes

Total soluble solids of crushed apple juice were determined using a hand held refrectometer (Master-M, Atago Co., LTD., Minato-Ku, Tokyo, Japan). The total soluble solids (TTS) were measured using the AOAC method 932.14 (AOAC 1998) [44]. The total titratable acidity of the cut apple cube was measured using the AOAC method 942.15 (AOAC 1998) [44]. A total of 10 g of apple cube was added to 100 g distilled water and blended, and then 0.3 mL of phenol phthalein indicator was added to it. The sample was then titrated with 0.1 N NaOH to a definite pink end-point.

### 3.8. Vitamin C Content of the Cut Apple Cubes

Vitamin C content of cut apple was measured by the AOAC method 985.33 (2,6-Dichloroindophenol titrimetric method, AOAC, 1998) [44]; 20 g of apple samples were blended with 50 mL of metaphosphoric acid–acetic acid solution. The homogenate was vacuum-filtered (vacuum pump-Rocker 300, Rocker Scientific Co., Ltd., New Taipei City, Taiwan) with Whatman No. 4 paper. A total of 10 mL of the filtered homogenate was titrated with a 2,6-dichloroindophenol standard solution. The indophenol solution was standardized by titrating sample blanks and an ascorbic acid standard solution (1 mg/mL). The titration volume was measured and used to quantify the vitamin C content of the cut apple (milligrams of ascorbic acid/g of sample, wet basis). Three repetitions for each treatment and two duplicates for each repetition were performed, throughout the experiment.

### 3.9. Microbiological Analysis

Ten grams of the cut apple samples were homogenized with 90 mL deionized water, using a homogenizer (CK-100, Chemist Scientific Corp. New Taipei City, Taiwan) and diluted with 0.1% peptone water, to obtain the microbial count. An appropriate dilution of 1.0 mL were made with 9 mL peptone water. Aerobic counts were determined on sterile disposable Petri dishes plate count agar (PDA, Merck, Darmstadt, Germany). They were incubated at 37 °C for 48 h. Molds and yeast were estimated on the potato dextrose agar (PDA, Merck, Darmstadt, Germany) and incubation conditions were 37 °C for 72 h. Each microbial count was obtained from the mean of three determinations and was expressed as log CFU/g (Song et al., 2013) [33].

### 3.10. Microscopic Examination of the Cut Apple Cubes

Microscopic examination was performed to evaluate the adherence to the apple cube surface and uniformity of the coating. Cross-sections of the coated cut apple cubes were observed using a Nikon Eclipse 50i optical microscope (Nikon H550S, Nikon Inc., Melville, NY, USA), coupled to a camera and a computer. The cut apple cubes were colored with a solution of toluidine blue to visualize and measure the coating thickness (Rojas-Grau et al., 2007) [31].

### 3.11. Sensory Evaluation

Forty-one graduate students of the Department of Food Science were instructed to evaluate the sour, sweet, and salty attributes through a triangle test and ranking test. A total of 0, 0.05%, 0.15%, and 0.25% citric acid solutions, 0, 0.15%, 0.3%, and 0.5% sucrose solutions, 0, 0.05%, and 0.25% salt solutions were served to the 41 participants; 5% milk was added to black tea with 5% sucrose and without sucrose for the triangle test. Panelists were required to be able to tell the differences in the milk tea and pass at least two ranking tests out of the three attributes. Cut apple cubes stored at 5 °C for 0, 4, 8, and 12 days were served to the 15 selected panelists, to evaluate color, odor, texture, flavor, and overall quality. A total of 7 male and 8 female graduate students of the Department of Food Science, between the ages of 21 and 32, were participants on the panel. Panelists were instructed to evaluate each attribute using a seven-point hedonic scale, ranging from “1 = dislike very much” to “7 = like very much”. Samples coded with three random digits were supplied to them. Each data point from sensory analysis represents an average of 15 panelists.

### 3.12. Statistical Analysis

Data was examined with an analysis of variance using the Statistics Package for Social Science statistics program (SPSS, 12, 1998). Duncan’s multiple range test was used to identify the difference between treatments at a 5% significance level (*p* < 0.05).

## 4. Conclusions

In our proposed edible coating system, fish gelatin, and a high molecular weight chitosan were used to examine the potential interactions between these molecules in the different experimental settings. This study was designed to understand the fish-gelatin–chitosan coating formation and its mitigation on deterioration of the physicochemical properties of cut apple cubes simulating the food processing parameters, at room temperature (22 °C) and at cold storage (5 °C). Chitosan provided a successful edible coating that extended shelf-life from 4 days up to 12 days at 22 °C, regardless of browning. Following this concept, the presence of fish gelatin with chitosan coating, improved the browning reaction, and vitamin C increased the cut apple cubes; the deterioration of the quality of the cut apple was slow down during the 12 days of storage at 5 °C. As a result, the high molecular weight of chitosan did mitigate the formation of enzymatic browning, and the firmness of the cut apple cubes, decreased at room temperatures. The shelf-life of chitosan-coated apple cubes could extended from 4 to 12 days, as per the total microbial counts. Decreased browning intensity of the cut apple cubes were achieved by fish-gelatin–chitosan coatings at cold storage. The fish-gelatin and chitosan mixtures showed an antibrowning effect at room temperature, and cold storage for 12 days; while chitosan alone showed microbial growth mitigation. The presence of the chitosan coating is desirable as it possesses antimicrobial activity, which could have potential edible film applications. The findings of the present study have further confirmed the beneficial properties of fish gelatin in mixtures of chitosan solution, which was found to be a viable approach for the mitigation of enzymatic browning in minimally processed apples, for preparing fresh-cut fruits and vegetables. Fresh-cut fruits have become attractive to consumers who are aware that healthy eating habits are important. However, it is difficult to preserve fresh-cut fruits for long-term storage. In this case, edible coatings and films are the proper way to maintain the quality of fresh fruits and vegetables, which could extend their shelf-life. Furthermore, using fish skin as the extraction of gelatin to produce biopolymer edible films, it might reduce the waste disposal issue of the fish-processing industry. Chitosan provided a successful edible coating that extended the shelf-life from 4 days up to 12 days, at 22 °C, regardless of browning. Following this concept, the presence of fish gelatin with chitosan coating, improved the browning reaction and increase the vitamin C content of the cut apple cubes, the deterioration of cut apple qualities had slowed down during the 12 days of storage at 5 °C. Decrease in the browning intensity of the cut apple cubes were achieved by fish-gelatin–chitosan coatings, at cold storage. The fish gelatin and chitosan mixtures showed an antibrowning effect at room temperature and at cold storage, for 12 days, while chitosan alone showed microbial growth mitigation. The presence of a chitosan coating is desirable, as it showed antimicrobial activity, which could have potential applications for edible films.

## Figures and Tables

**Figure 1 molecules-24-02008-f001:**
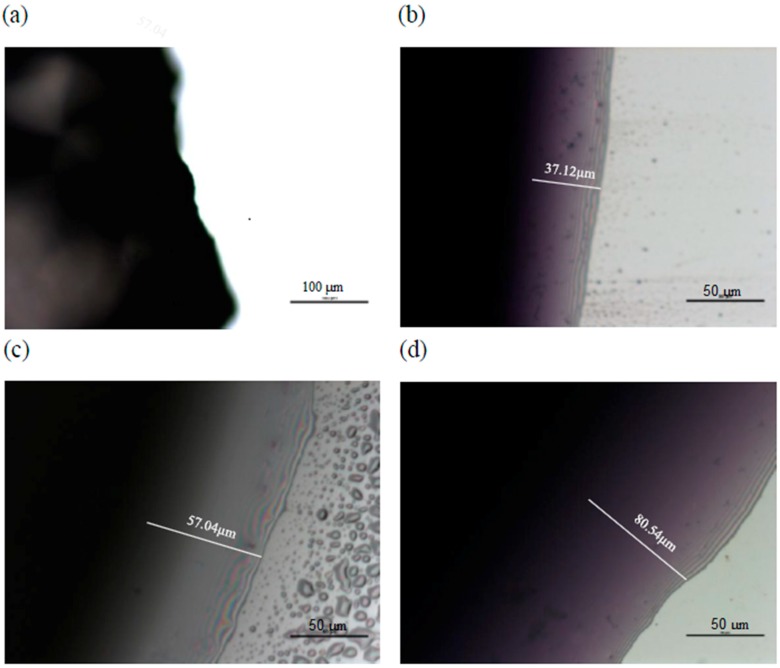
Micrograph of an apple (cross-section) coated with (**a**) control; (**b**) chitosan; (**c**) gelatin, and (**d**) chitosan–gelatin. The scale bar represents (**a**) 100 µm and (**b**–**d**) 50 µm.

**Figure 2 molecules-24-02008-f002:**
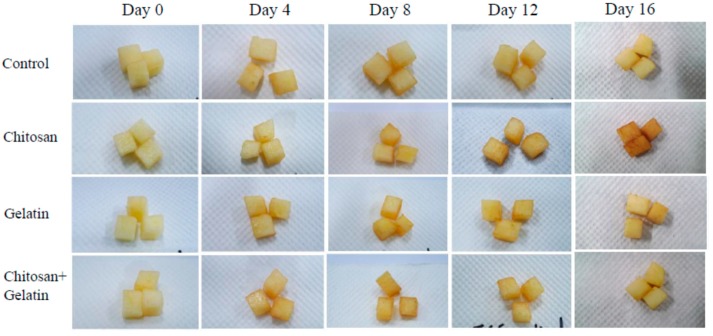
Effect of different coatings on the appearance of fresh-cut apple during storage for 16 days at 5 °C.

**Figure 3 molecules-24-02008-f003:**
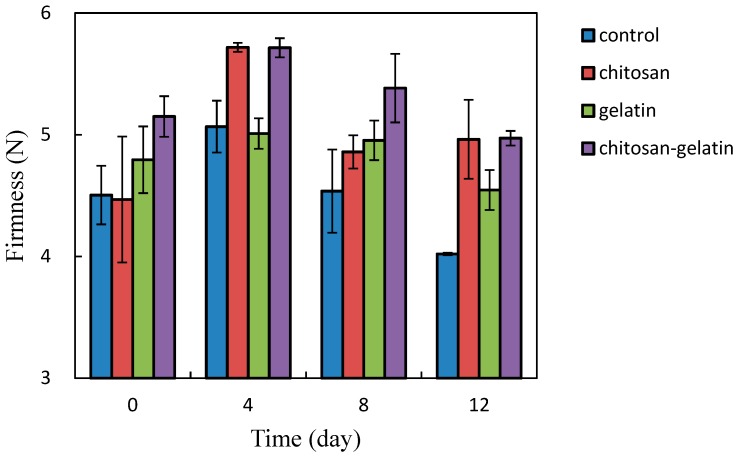
Effect of the different coatings on the firmness of fresh-cut apple during 12 days of storage at 5 °C. Data are the mean ± SD.

**Table 1 molecules-24-02008-t001:** *L** value changes of cut apple cubes coated with different coatings and stored at 5 °C and 22 °C for 12 days.

*L** Value	Sample Treatment	Storage Time (Day)			
		0	4	8	12
22 °C	Control	68.23 ± 0.41 ^A^	51.22 ± 1.12 ^B^	- *	-
	Chitosan	65.29 ± 0.01 ^A^	49.75 ± 0.00 ^B^	39.46 ± 0.00 ^C^	42.02 ± 0.38 ^C^
	Gelatin	67.26 ± 0.72 ^A^	44.49 ± 1.86 ^C^	-	-
	Chitosan+Gelatin	65.92 ± 1.03 ^A^	48.60 ± 3.01 ^B^	-	-
5 °C	Control	67.70 ± 1.32 ^A^	63.58 ± 0.86 ^A^	64.00 ± 0.78 ^A^	64.90 ± 1.73 ^A^
	Chitosan	67.84 ± 1.78 ^A^	62.08 ± 3.00 ^A^	62.39 ± 1.86 ^A^	55.98 ± 0.50 ^B^
	Gelatin	67.51 ± 1.60 ^A^	59.19 ± 3.47 ^A^	57.30 ± 0.85 ^B^	54.41 ± 1.70 ^B^
	Chitosan+Gelatin	65.05 ± 0.56 ^A^	60.39 ± 1.31 ^A^	60.41 ± 0.64 ^A^	62.92 ± 1.79 ^A^

Values in a column (A to C) followed by different superscript letters are significantly different (*p* < 0.05). * showed decayed.

**Table 2 molecules-24-02008-t002:** Browning index of cut apple cubes coated with different coatings and stored at 5 °C and 22 °C for 12 days.

Browning Index	Sample Treatment	0	4	8	12
22 °C	Control	80.84 ± 0.18 ^AB^	95.74 ± 2.39 ^AB^	- *	-
	Chitosan	87.13 ± 4.48 ^AB^	86.03 ± 6.10 ^AB^	80.82 ± 17.46 ^A^	83.10 ± 4.11 ^A^
	Gelatin	112.98 ± 5.32 ^A^	119.76 ± 3.36 ^A^	-	-
	Chitosan+Gelatin	85.79 ± 0.89 ^AB^	93.74 ± 9.72 ^AB^	-	-
5 °C	Control	88.29 ± 1.64 ^AB^	94.66 ± 6.52 ^AB^	89.02 ± 5.86 ^A^	86.87 ± 1.23 ^A^
	Chitosan	75.98 ± 0.07 ^B^	90.32 ± 0.00 ^B^	88.18 ± 0.00 ^A^	82.53 ± 4.35 ^A^
	Gelatin	78.18 ± 2.46 ^AB^	91.81 ± 3.13 ^B^	88.71 ± 3.66 ^A^	96.95 ± 6.62 ^A^
	Chitosan+Gelatin	80.03 ± 1.22 ^AB^	89.95 ± 7.08 ^B^	87.56 ± 0.96 ^A^	82.32 ± 1.40 ^A^

Values in a column (A to B) followed by different superscript letters are significantly different (*p* < 0.05). * showed decayed.

**Table 3 molecules-24-02008-t003:** pH changes of cut apple cubes coated with different coatings and stored at 5 °C and 22 °C for 12 days.

pH	Sample Treatment	Storage Time (Day)			
		0	4	8	12
22 °C	Control	3.83 ± 0.24 ^A^	3.86 ± 0.18 ^BC^	- *	-
	Chitosan	4.18 ± 0.15 ^A^	4.59 ± 0.19 ^AC^	4.64 ± 0.23 ^A^	4.86 ± 0.02 ^AD^
	Gelatin	3.94 ± 0.27 ^A^	3.89 ± 0.21 ^BC^	-	-
	Chitosan+Gelatin	4.06 ± 0.03 ^A^	4.06 ± 0.21 ^AB^	-	-
5 °C	Control	3.86 ± 0.24 ^A^	4.07 ± 0.32 ^AB^	3.96 ± 0.45 ^A^	4.18 ± 0.13 ^BC^
	Chitosan	4.21 ± 0.31 ^A^	4.36 ± 0.37 ^AB^	4.52 ± 0.27 ^A^	4.48 ± 0.23 ^BD^
	Gelatin	3.88 ± 0.29 ^A^	4.01 ± 0.37 ^AB^	3.89 ± 0.10 ^A^	4.04 ± 0.06 ^CD^
	Chitosan+Gelatin	4.04 ± 0.30 ^A^	4.15 ± 0.31 ^AB^	4.04 ± 0.41 ^A^	4.31 ± 0.25 ^BC^

Values in a column (A to D) followed by different superscript letters are significantly different (*p* < 0.05). * showed decayed.

**Table 4 molecules-24-02008-t004:** Total soluble solids (TSS) changes of cut apple cubes coated with different coatings and stored at 5 °C and 22 °C for 12 days.

TSS (°Brix)	Sample Treatment	Storage Time (Day)			
		0	4	8	12
22 °C	Control	10.5 ± 1.0 ^A^	10.8 ± 0.4 ^AC^	- *	-
	Chitosan	10.6 ± 0.8 ^A^	10.8 ± 0.1 ^AC^	10.7 ± 0.0 ^A^	10.5 ± 0.1 ^AD^
	Gelatin	10.3 ± 0.8 ^A^	11.5 ± 0.1 ^AC^	-	-
	Chitosan+Gelatin	11.0 ± 1.6 ^A^	10.4 ± 0.6 ^AA^	-	-
5 °C	Control	11.1 ± 1.1 ^A^	10.8 ± 0.0 ^A^	11.2 ± 0.7 ^A^	11.4 ± 0.5 ^AA^
	Chitosan	10.6 ± 0.8 ^A^	10.8 ± 0.4 ^A^	10.6 ± 1.2 ^A^	10.2 ± 0.6 ^AD^
	Gelatin	10.1 ± 1.6 ^A^	10.9 ± 0.4 ^A^	10.8 ± 0.7 ^A^	10.6 ± 0.2 ^AD^
	Chitosan+Gelatin	10.6 ± 1.5 ^A^	10.4 ± 1.3 ^A^	10.6 ± 0.6 ^A^	10.9 ± 0.2 ^AA^

Values in a column (A to D) followed by different superscript letters are significantly different (*p* < 0.05). * showed decayed.

**Table 5 molecules-24-02008-t005:** Change in the populations of total aerobic bacteria, yeast, mold, and coliform in fresh-cut apple slices stored 12 days at 5 °C.

Microbiological Analysis	Sample Treatment	Storage Time (Day)			
		0	4	8	12
Total aerobic bacteria	Control	<1	<1	<1	4.59 ^Aa^
	Chitosan	<1	<1	<1	<1
	Gelatin	<1	1.92 ^Ab^	2.23 ^Ab^	5.34 ^Aa^
	Chitosan+Gelatin	<1	<1	<1	<1
Yeast and mold	Control	<1	<1	2.36 ^Ab^	5.02 ^Aa^
	Chitosan	<1	<1	<1	<1
	Gelatin	<1	<1	3.03 ^Ab^	5.64 ^Aa^
	Chitosan+Gelatin	<1	<1	<1	<1
Coliform	Control	<1	<1	<1	4.81 ^Aa^
	Chitosan	<1	<1	<1	<1
	Gelatin	<1	2.55 ^Aab^	2.72 ^Aab^	5.30 ^Aa^
	Chitosan+Gelatin	<1	<1	<1	<1

Values in a row (a to b) or a column (A to B) followed by different superscript letters are significantly different (*p* < 0.05).

**Table 6 molecules-24-02008-t006:** Effect of the different coatings on the sensory scores of fresh-cut apple, during 12 days of storage at 5 °C.

Sensory Scores	Sample Treatment	Storage Time (Day)			
		0	4	8	12
Color	Control	4.15 ± 1.28 ^Ba^	3.00 ± 1.00 ^Ab^	2.87 ± 1.06 ^Cb^	2.93 ± 1.33 ^BbA^
	Chitosan	4.38 ± 1.12 ^Ba^	3.77 ± 1.30 ^Aa^	4.60 ± 0.99 ^Aa^	4.14 ± 1.56 ^AaA^
	Gelatin	5.77 ± 1.17 ^Aa^	3.92 ± 1.55 ^Ab^	3.87 ± 1.25 ^ABb^	3.27 ± 1.10 ^ABb^
	Chitosan + Gelatin	3.85 ± 1.46 ^Ba^	3.62 ± 1.56 ^Aa^	3.33 ± 1.11 ^BCa^	4.07 ± 1.00 ^AaA^
Odor	Control	4.92 ± 1.26 ^Aa^	4.31 ± 1.25 ^Aaa^	4.13 ± 1.25 ^AaA^	4.00 ± 1.41 ^AaA^
	Chitosan	3.46 ± 1.27 ^Ba^	3.62 ± 1.19 ^Aaa^	3.00 ± 0.93 ^Bab^	2.57 ± 0.76 ^BbA^
	Gelatin	5.54 ± 1.27 ^Aa^	4.54 ± 0.97 ^Aab^	3.87 ± 1.13 ^ABb^	4.33 ± 1.76 ^AbA^
	Chitosan + Gelatin	3.54 ± 1.13 ^Ba^	3.62 ± 0.96 ^Aaa^	3.33 ± 1.23 ^ABa^	3.50 ± 1.29 ^ABa^
Texture	Control	3.38 ± 1.45 ^Aa^	2.92 ± 1.12 ^Baa^	3.00 ± 1.00 ^Ba^	3.29 ± 1.54 ^Baa^
	Chitosan	3.77 ± 0.93 ^Ab^	4.00 ± 1.41 ^ABab^	4.80 ± 0.86 ^Aa^	4.57 ± 0.76 ^Aab^
	Gelatin	3.77 ± 1.83 ^Aa^	3.62 ± 1.33 ^ABaa^	3.33 ± 1.23 ^Ba^	3.27 ± 1.10 ^Baa^
	Chitosan + Gelatin	3.77 ± 1.83 ^Aa^	3.62 ± 1.33 ^ABaa^	3.33 ± 1.23 ^Ba^	3.27 ± 1.10 ^Baa^
Flavor	Control	4.15 ± 1.68 ^Aa^	3.46 ± 1.05 ^Aaa^	4.27 ± 0.96 ^Aaa^	3.29 ± 1.68 ^ABaa^
	Chitosan	4.31 ± 1.25 ^Aa^	3.54 ± 1.27 ^Aab^	3.60 ± 0.99 ^ABab^	3.14 ± 1.23 ^BbaA^
	Gelatin	4.46 ± 1.56 ^Aa^	3.54 ± 0.97 ^Aaa^	4.13 ± 1.64 ^ABa^	4.20 ± 0.86 ^AaaA^
	Chitosan + Gelatin	4.38 ± 1.56 ^Aa^	3.77 ± 1.01 ^Aab^	3.27 ± 1.16 ^Bba^	3.86 ± 1.03 ^ABab^
Overall quality	Control	3.69 ± 1.18 ^Aa^	3.31 ± 0.95 ^Baa^	3.87 ± 0.64 ^ABa^	3.21 ± 1.37 ^Aaa^
	Chitosan	3.92 ± 1.26 ^Aa^	4.00 ± 1.00 ^ABa^	4.07 ± 0.70 ^AaA^	3.29 ± 0.83 ^Aaa^
	Gelatin	4.23 ± 1.30 ^Aa^	3.77 ± 1.17 ^ABa^	4.13 ± 1.41 ^AaA^	4.00 ± 0.93 ^Aaa^
	Chitosan + Gelatin	4.15 ± 1.21 ^Aa^	4.23 ± 0.83 ^Aaa^	3.27 ± 1.10 ^BbA^	3.79 ± 1.05 ^Aab^

Data are mean ± standard deviation (n = 15). Values in a row (a to b) or a column (A to C) followed by different superscript letters are significantly different (*p* < 0.05). Score 7 = like extremely; 4 = neither like nor dislike; 1 = dislike extremely.

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
