# Peer review of "Effect of Chitosan and Fish Gelatin Coatings on Preventing the Deterioration and Preserving the Quality of Fresh-Cut Apples"

_molecules, 2019, doi:10.3390/molecules24102008_

Reviewer 1 Report

The manuscript studies the use of chitosan and gelatin fish to protect fruit pieces. The paper in general is well written and clear. The subject have been well studied previously. The references must be updated, there is a lot of new papers in the last years using the gels to protect fruits. So it is important in this paper to show clearly the innovation presented.

Some considerations:

It must be well explain how the biomaterial was applied to the fruit cubes. The industrial applicability is very important in this kind of processes. How justify that the fruits is cut and maintain in the freeze (it is clear that at 22 ºC the effect is not sufficiently good) at 5 ºC.

Another important consideration is the analysis if the process could be economically possible. Some comments could be included in the discussion part.

I think the paper is well presented, the techniques well used but the results must be more discussed, specially the real applicability of the application of the chitosan gels in the fruits.

Author Response

Dear Reviewer

The authors are extremely grateful to two anonymous referee involved for providing his/her excellent comments and valuable advice in this paper. We have revised the paper based on the referee’s comments. We have pleasure in requesting the referee to review this paper. Thank you. Your prompt attention to this paper will be much appreciated.

Point 1:

The manuscript studies the use of chitosan and gelatin fish to protect fruit pieces. The paper in general is well written and clear. The subject has been well studied previously. The references must be updated, there is a lot of new papers in the last years using the gels to protect fruits. So it is important in this paper to show clearly the innovation presented.

Response 1: 

Thank you, we have updated 14 references related to this topic as marked reference in revised manuscript. The innovation presented in this paper is we use fish gelatin instead of pork gelatin for certain religious groups.

Point 2: 

Some considerations: It must be well explain how the biomaterial was applied to the fruit cubes. The industrial applicability is very important in this kind of processes. How justify that the fruits is cut and maintain in the freeze (it is clear that at 22℃ the effect is not sufficiently good) at 5℃.

Response 2:

Fresh-cut apples have attracted increasing attention because of convenience and fresh-like character in convenient store. However, fresh-cut apples are more perishable and susceptible to deterioration accompanied with microbial, surface browning, softening and water loss. When applied chitosan to fresh-cut apples, it shows triple activity: 1 film formation on the wound surface; 2 antimicrobial activity; and 3 improvement of host defenses. Fresh-cut apples are recommended to kept at 5 to reduce the rate of deterioration and biochemical reaction. We also kept the fresh-cut apples at room temperature (22℃) to evaluate how fast they deteriorated.

Point 3: 

Another important consideration is the analysis if the process could be economically possible. Some comments could be included in discussion part.

Response 3: 

The process is economical because the cost of fish gelatin and chitosan is not expensive and easy to get.

Point 4: 

I think the paper is well presented, the techniques well used but the results must be more discussed, specially the real applicability of the application of the chitosan gels in the fruits.

Response 4:

Thanks for the comments. We have revised and added some discussion in the revised manuscript as the red marked text.

Thank you for your consideration.

Yours truly,

Wen-Chieh Sung, Ph.D.

Professor

Department of Food Science

National Taiwan Ocean University

Reviewer 2 Report

See file attached

Author Response

Dear Reviewer

The authors are extremely grateful to two anonymous referee involved for providing his/her excellent comments and valuable advice in this paper. We have revised the paper based on the referee’s comments. We have pleasure in requesting the referee to review this paper. Thank you. Your prompt attention to this paper will be much appreciated.

Reviewer #2

Point 1:

See file (peer_review-4280691.v2.pdf) attached.

.

Response 1: 

Thanks for your assistance and comments in our manuscript. We have revised the manuscript according to your comments and suggestions. Please see the revised manuscript as the attached file.

Point 2: 

Insert the correct scientific name and cultivar used in this study Insert the correct scientific name and cultivar used in this.

Response 2:

We check and change the scientific name and cultivar of apples in abstract as the revised manuscript. Thanks for pointing out the mistake.

Point 3: 

Improve this sentence in abstract in lines 15 and 16.

Response 3: 

We revise the sentence in lines 15-16 to “Chitosan provided an effective control in microbial growth, maintained firmness during 4 days of storage at room temperature (22) and 12 days at refrigerator (5).”

Point 4: 

Improve this sentence in introduction section in lines 28-31 at page 1. Fruit browning is due to several factors.

Response 4:

Thanks for the valuable comments again. We can feel the referee spent so much time reading our manuscript very carefully. We have revised the sentence in lines 28-31 at page 1 and added several sentences “Fresh-cut apples are recommended to keep at 5 to reduce the rate of deterioration and biochemical reaction (Gil et al., 2006).  Fresh-cut fruits are not easy to preserve color than are whole fruits due to the cutting operation causes a release of polyphenol oxidase, which reacts with pehenolic compounds and forms surface browning (Albanese et al., 2007; Pilon et al., 2014). Different apple varieties, pre- and postharvest factors such as concentration of gases in the storage atmosphere influence their susceptibility to browning (Hatoum et al., 2014).” in the first paragraph. Please give us suggestion and hope this revised introduction could have a great improvement.

Point 5: 

The Authors should check in the whole manuscript the word "fruits".

Response 5:

We check and change the word “fruit” to “fruits” at page 1 lines 22 and 23, page 2 line 47 as the red marked text. Thanks for pointing out the mistakes again.

Point 6: 

Insert time of storage.

Response 6:

The sentence in the last paragraph of introduction was revised to “The main objective of this research was to evaluate the effect of fish gelatin and chitosan edible coatings on the quality of fresh-cut apples during storage at 5°C for 12 days and 22°C for 4 days.”

Point 7: 

The Authors should indicate the statistic differences among treated and untreated samples.

Response 7:

The sentence in the last sentence of 2.1 Weight loss of cut apple cubes section was revised to “Coating did not significantly reduce the weight loss in cut apple cubes, although weight loss increased during storage in this study, the weight loss (2.07%) of the fish gelatin-chitosan coated apple cubes was slightly lower (p > 0.05) after 12 days of storage at 5°C compared to the control (2.53%) and other tested groups.” They are not statistic differences among treated and untreated samples.

Point 8: 

Check this word “smples”.

Response 8:

We revised the word “smples” to “samples” at page 3 line 117 as the red marked text. Sorry for the mistake again.

Point 9: 

Improve this sentence.

Response 9:

We revised the sentence to “Application of a combined fish gelati-chitosan coating is an effective approach to inhibit enzymatic browning and microbial growth of cut apple cubes for 12 days at 5°C (Table 2).” at page 4 lines 132-134 as the red marked sentence. Thank you for your valuable suggestion.

Point 10: 

Check this mistake in all tables.

Response 10:

We revised the word “chitoan” to “chitosan” at pages 5, 6, 9, 11, 13 & 15 from Table 1 to Table 6 as the red marked text. Sorry for the serve mistakes again.

Point 11: 

The Authors should insert the letters in each time for different treatments in the Figure 3.

Response 11:

Thanks for the comment again. We have revised the Figure 3 and added letters to show the difference among different treatments at page 8 and marked at different color in Figure 3. Please give us suggestion and hope this revised Figure 3 could have a great improvement.

Point 12: 

Check the values reported in this sentence. They are not in agreement with Table 3. The Authors should indicate statistical analysis "statistic significant or not".

Response 12:

We revised the paragraph of section 2.5 pH of cut apple cubes to “A slight increase in pH values was observed after 12 days storage at 5°C (p > 0.05) (Table 3). However, a significant increase in pH value (p < 0.05) of chitosan coated cut apple cubes from pH 4.18 to pH 4.86 was measured during 12 days storage at 22°C (Table 3) comparing to the other groups which the pH of cut apple cubes did not change dramatically during storage at 22°C for 4 days (Table 3).” at page 8. Hope it has a great improvement.

Point 13: 

This sentence is not in agreement with the data reported in Table 4.

Response 13:

We are very sorry for the mistake. We added the word “no” to the first sentence “There was no …” at page 10 as the red marked text. Sorry for the serve mistake again.

Thank you for your consideration.

Yours truly,

Wen-Chieh Sung, Ph.D.

Professor

Department of Food Science

National Taiwan Ocean University

Round  2

Reviewer 1 Report

The authors have corrected the paper adequately.

Author Response

Dear Reviewer

The authors are extremely grateful to two anonymous referee involved for providing his/her excellent comments and valuable advice in this paper. We have revised the paper based on the referee’s comments. We have pleasure in requesting the referee to review this paper. Thank you. Your prompt attention to this paper will be much appreciated.

Reviewer #1

Point 1:

The authors have corrected the paper adequately.

Response 1: 

Thanks for all the valuable comments and suggesting update the references and revising the manuscript. We have checked spell in the revised manuscript as the red marked text.

Yours truly,

Wen-Chieh Sung, Ph.D.

Professor

Department of Food Science

National Taiwan Ocean University

Reviewer 2 Report

Figure 3 The Authors should check  statistical analysis.

Author Response

Dear Reviewer

The authors are extremely grateful to two anonymous referee involved for providing his/her excellent comments and valuable advice in this paper. We have revised the paper based on the referee’s comments. We have pleasure in requesting the referee to review this paper. Thank you. Your prompt attention to this paper will be much appreciated.

Reviewer #2

Point 1:

Figure 3 The Authors should check statistical analysis.

Response 1: 

Thanks for your assistance and comments in our manuscript. We have checked statistical analysis of the Figure 3 and revised it as bar figure. Please see the revised manuscript.

Thank you for your consideration.

Yours truly,

Wen-Chieh Sung, Ph.D.

Professor

Department of Food Science

National Taiwan Ocean University